# Calibration of Steel Rings for the Measurement of Strain and Shrinkage Stress for Cement-Based Composites

**DOI:** 10.3390/ma13132963

**Published:** 2020-07-02

**Authors:** Adam Zieliński, Maria Kaszyńska

**Affiliations:** Faculty of Civil Engineering and Architecture, West Pomeranian University of Technology in Szczecin, al. Piastów 50, 70-311 Szczecin, Poland; maria.kaszynska@zut.edu.pl

**Keywords:** restrained ring test, autogenous shrinkage cracking, concrete cracking test, concrete shrinkage cracking test, restrined ring calibration

## Abstract

Concrete shrinkage is a phenomenon that results in a decrease of volume in the composite material during the curing period. The method for determining the effects of restrained shrinkage is described in Standard ASTM C 1581/C 1581M–09a. This article shows the calibration of measuring rings with respect to the theory of elasticity and the analysis of the relationship of steel ring deformation to high-performance concrete tensile stress as a function of time. Steel rings equipped with strain gauges are used for measurement of the strain during the compression of the samples. The strain is caused by the shrinkage of the concrete ring specimen that tightens around steel rings. The method allows registering the changes to the shrinkage process in time and evaluating the susceptibility of concrete to cracking. However, the standard does not focus on the details of the mechanical design of the test bench. To acquire accurate measurements, the test bench needs to be calibrated. Measurement errors may be caused by an improper, uneven installation of strain gauges, imprecise geometry of the steel measuring rings, or incorrect equipment settings. The calibration method makes it possible to determine the stress in a concrete sample leading to its cracking at specific deformation of the steel ring.

## 1. Introduction

The shrinkage of composite materials is a phenomenon where the material reduces its volume as a result of drying, carbonation, and autogenous processes [1,2,3,4,5]. If an element is not restrained and can freely change its volume, the structure remains intact. However, when the shrinkage is restrained, the lack of free strain results in the development of internal stresses that lead to cracking.

One of the basic research methods for the controlled reduction of concrete shrinkage deformations is the use of ring methods. Presumably, the first tests of this type were carried out by Carlson and Reading [6] in the 1940s, where the result of the research was the age of cracking of concrete ring samples. The geometry and cross-section of the concrete ring can be selected based on the size of the aggregate. The degree of limitation depends on the modulus of elasticity and width of the two rings: the concrete ring and the rigid steel ring limiting the free deformability of the composite. However, height is a generally accepted parameter. Different geometries of limiting rings [7,8,9] and annular concrete samples [7,10,11,12] were developed. Two steel measuring rings were used: external and internal, where an additional external ring was used to limit deformations caused by autogenous swelling and the thermal expansion of concrete [13]. Studies on elliptical rings have been implemented to achieve earlier concrete cracking [14,15]. Two standards for ring tests have been developed in the USA: the AASHTO bridge standard T 334-08 and ASTM 1581M–09a.

Established in the standard ASTM C 1581/C 1581M–09a “Determining Age at Cracking and Induced Tensile Stress Characteristics of Mortar and Concrete under Restrained Shrinkage” dimensions of the steel and concrete rings mean that the tensile stresses due to the constraints are similar to the tensile stresses due to the drying of the outer surface of concrete samples. This configuration of boundary stress causes uniform straining of the concrete section. A similar value of the edge tensile stress determines the fracture of the concrete sample as a result of exceeding its tensile strength [16]. The rings method uses strain gauge measurement of the steel ring strain caused by the shrinkage of concrete. The significant advantage of this method is that the recording of strains starts right after the sample is formed.

In modern concretes with low water/cement ratios, the overall shrinkage is significantly affected by the autogenous shrinkage, which occurs in the first stage of hardening. High-performance concretes undergo autogenous shrinkage even up to 200 µm/m after the first day of maturing. In the case of traditional concretes with a water/cement ratio of 0.5, the value of autogenic shrinkage after 28 days reaches 100 µm/m and in practical conditions is negligible [1]. Cracking caused by the shrinkage increases the penetration depth of water and aggressive substances that cause the corrosion of rebar, concrete leaching, and as a result, the deterioration of concrete’s durability and structural failure. So far, a lot of research has been done to improve durability and minimize concrete susceptibility to cracking. The studies analyzed the impact of changing climatic conditions affecting the fracture rate of concrete samples [10,17,18] and the rate at which drying begins [19,20]. The effect of concrete composition on cracking susceptibility was also investigated [7,9,21,22,23]. The research also included the effect of internal curing soaked aggregate [24,25], fibers [7,8,9,26,27], admixtures reducing shrinkage [28,29]. Numerical simulation tests were also performed in predicting concrete susceptibility to cracking based on ring methods [30,31,32].

Tests performed in accordance with the ASTM C 1581/C 1581M–09a standard allow determining concrete sample cracking time as a result of restrained shrinkage exceeding concrete tensile strength. However, it is not possible to determine the exact value of the shrinkage; instead, the strain of the steel ring needs to be measured. Before test measurements can be used in further analysis, the steel measuring rings must be calibrated. The calibration process eliminates measurement errors caused by strain gauge installation, which could give different results than those calculated with theoretical equations. Those errors can significantly affect or even disrupt the mesurements entirely. Tests performed on calibrated steel rings using the restrained ring method allow accurately measuring strains in steel rings and make it possible to determine tensile stresses in concrete ring samples.

The article presents the calibration process of three steel measuring rings. Using calibrated restrained rings, the testing procedure was carried out for two self-compacting high-performance concretes with light and natural aggregate. Obtained steel ring deformation values and developed tensile stresses in annular concrete samples were analyzed for two maturation conditions: deformation due to autogenous and drying shrinkage—the side formwork removed after 24 hours of concreting—and deformation due to autogenous shrinkage only without side surface drying effects. The use of various test modes has made it possible to check the measurement precision and stability of strain development during short- and long-term tests.

## 2. Research Problem

The aim of the study was to calibrate three steel measuring rings for deformation registration in accordance with values resulting from the theory of elasticity. A novelty of this test is the calibration stand and procedure dedicated to measuring steel rings strain according to ASTM C 158/C 1581M–09a, which obtained a patent for an invention.

## 3. Methods and Experiment Program 

### 3.1. Description of the Test Bench

The basic scheme of the calibration test bench is shown in Figure 1. The steel measuring ring equipped with strain gauges installed circumferentially on the internal surface must be set in the center of the outer shielding ring and fixed to the bottom plate. To apply external pressure for calibration, a rubber inflatable collar should be placed in between the measuring ring and the outer shielding. Then, the rings should be covered with a rigid top plate. Bottom and top plates should be made of a non-deformable material such as steel and fixed to each other with bolts. The outer ring should be 5 mm higher than the measurement ring to allow free deformation. Such a design of the test bench allows for the application of compressive stresses on the inner measuring ring from fixed outer shielding and fixed horizontal plates.

The rubber collar should be connected through a digital manometer to the air compressor for simultaneous recording of its pressure and the deformation of the measuring ring. The steel measuring ring is connected with cables to strain gauge bridge and measurement equipment. The calibration system shown in Figure 1 uses a strain gauge bridge with internal temperature compensation. 

The system used in the lab ulitizes a strain gauge bridge without the internal temperature compensation, which requires connecting strain gauges with Wheatstone half- or full-bridge circuits. Each measuring point consisted of a pair of strain gauges, which were vertically and annularly glued to the inner surface of the steel ring. The temperature compensation was provided by the strain gauges placed in the vertical axis, which are a part of a circuit of another measuring ring. The setup is shown in Figure 2 and a block diagram is presented in Figure 3. Calibration was carried out for three measuring rings, with four pairs of strain gauges spaced every 90 degrees. Strain gauges were installed in the circumferential direction, halfway up the inner surface of the steel rings. To compensate the temperature impact, recordings were taken from strain gauges installed in an additional measuring ring that was not actively involved in calibration, as shown in Figure 4a.

### 3.2. Experimental Procedure

First, the passive stage of the calibration begins with placing the measuring ring on the test bench, connecting it tightly to the plates, and connecting the measuring equipment and the compressor. The active calibration process starts in the second stage, as shown in Figure 3. Air pumped by the compressor passed through the hose with a digital manometer to the collar. Once the space between the ring and shield plate fills, the collar starts to impose even radial pressure on the surrounding surfaces, including the external surface of the steel ring. The strain gauges register the change in resistance and send the impulse to the gauge bridge responsible for calculating the strain of the steel ring. From the gauge bridge, the signal is sent to the computer, which shows the measurements as a continuous graph of ring strain function. Figure 4 shows the test bench during the ring calibration process.

Additionally, to minimize friction between the expanding collar and the measuring ring, the outer surfaces of the measuring ring, collar, and inner surface of the outer ring were covered with synthetic oil before the test. The friction of expanding torus on the outer surface of the measuring ring can cause discrepancies and uneven strain. This results from Poisson’s ratio for steel and can induce cumulative measurement error for each calibration of measuring rings.

The measurements allow acquiring a time function of pressure and strain. The result is visible as a linear dependency between circumferential strain and radial stress. A comparison of functions, which were both acquired from the measurements and calculated from theoretical equations, allows determining the calibration coefficient for the tested ring. The calibration allows comparing the results of measured strains for three independent rings. 

Calibration analysis was made individually for three steel rings, using a theoretical function [6]:(1)$$\sigma_{R}=-\varepsilon_{\theta }\cdot E_{s}\cdot \frac{r_{os}{}^{2}-r_{is}{}^{2}}{2r_{os}{}^{2}}$$
where $\sigma_{R}$ represents the external pressure imposed on the steel ring (MPa),$\;\varepsilon_{\theta }$ represents the circumferential strain of the steel ring (m/m·10^−3^), $E_{s}$ represents the elasticity modulus of the steel ring (GPa),$\;r_{os}$ represents the outer radius of the steel ring (mm), and $r_{is}$ represents the inner radius of the steel ring (mm).

Based on the calibrated relationship of circumferential deformation of measuring rings $\varepsilon_{\theta }$, to the value of radial pressure $\sigma_{R}$, the peripheral stress course in concrete ring samples is determined. The largest value of peripheral stresses in the concrete sample is recorded in the nearest zone of the radial stress of the steel ring—on the inner surface of the concrete sample [6]:(2)$$\sigma_{\theta max,c}=\sigma_{R}\cdot \left(\frac{r_{oc}{}^{2}}{r_{ic}{}^{2}}+1\right)/\left(\frac{r_{oc}{}^{2}}{r_{ic}{}^{2}}-1\right)$$
where $\sigma_{\theta max,c}$ represents the maximum circumferential stress in concrete specimen (MPa),$\;r_{oc}$ represents the outer radius of the concrete specimen (mm), and $r_{ic}$ represents the inner radius of the concrete specimen (mm).

## 4. Results

### 4.1. Calibration Test Results

The steel ring deformations were recorded individually for each of the four circumferential strain gauges as a function of time and depending on the acting pressure. To eliminate potential measurement errors and to increase the precision of the calibration, the measurement of the pressure acting on each ring and the measurement of strain at each strain gauge was taken 6 times. This allowed incorporating three measuring cycles, turning the steel ring around the rubber collar each time, with two measurements per cycle. Then, the mean value of the steel ring strain could be calculated. The influence of air pressure that ranged from 0 to 5.5 bars on the strain function in time was consistent and repeatable for each tested ring, as shown in Figure 5. 

The measured values of steel ring strains per each gague and for each trial test are shown in Table 1. The table also shows the average strains per each gauge from all trials and the average strain for the whole ring per each trial. 

Based on the deviations shown in Table 1, it can be observed that the strain gauges of rings A and B were installed properly and the geometry of the ring is within 2%. It is assumed that deviation of up to 5% accounts for fabrication imprefections, and its impact is negligible. Deviations above 5% and up to 15% require the application of a calibration factor, which is derived and applied to an individual strain gauge or to the whole ring. Larger strain deviation requires elimination of the measuring ring from the tests. In such a situation, it is necessary to remove faulty strain gauges and verify ring geometry.

Table 1 also shows that for ring C, the measured values differed by 6% from the theoretical ring model. Circumferential strain gauges No. 1, 2, and 4 on rings A, B, and C record similar strain values, while strain gauge No. 3 on ring C shows value lower than the values for the corresponding strain gauge on rings A and B. This indicates a poor installation of the third circumferential strain gauge and a correct geometry of the steel ring. As mentioned above, for ring C, a calibration coefficient has to be applied due to the measured strain divergance off the theoretical values within 15%. Tolerance ranges between ±5% and ±15% were analyzed for each circumferential strain gauge and for the mean ring deformation value relative to the theoretical value.

When strain gauges record differenciated strain values at a constant pressure level, this indicates their incorrect or non-parallel installation on the inner surface of the ring. However, if all recorded deformation values are similar and lower or higher than the theoretical value, then most likely, the measuring ring geometry differs. 

Figure 6 shows the measurement accuracy of the tested rings relative to the theoretical strain values. Rings A and B show strain values close to the ones calculated from Equation (1), whereas ring C had an extensive measurement error.

Figure 7 shows the measured function of circumferential strain–radial stresses for steel rings and the theoretical curve. The strain function for rings A and B develops in accordance to the theoretical relationship. Based on this, it can be stated that rings A and B are calibrated properly, and there is no need for additional amplification through a calibration factor. The strains of ring C differ significantly from theoretical calculations. To properly calibrate ring C, it is necessary to change the slope coefficient of the circumferential strain–radial stress function.

### 4.2. Calibration Coefficient for Individual Ring

The result of calibration process is an individually determined ring calibration coefficient (3) that adjusts the slope coefficient of the measured value plot to the theoretical plot. This coefficient accounts for ring geometrical imperfections and faulty strain gauge installation. The calibration coefficients for the three measuring rings considered are shown in Table 2.
(3)$$\gamma_{c}=\frac{\varepsilon_{\theta .t}}{\varepsilon_{\theta .m}}$$
where $\varepsilon_{\theta .t}$ represents the theoretical circumferential strain of the steel ring at given pressure (m/m·10^−6^), $\varepsilon_{\theta .m}$ represents the measured circumferential strain of the steel ring at given pressure (m/m·10^−6^), and $\gamma_{c}$ represents the calibration coefficient.

The recorded strains of rings A and B are within the lower bound tolerance of 5%, so they do not need to be calibrated, and they can be directly used in further analyses. The measured strains for ring C must be computed, including the calibration coefficient, accordingly to the equation:(4)$$\varepsilon_{\theta .n}=\gamma_{c}\cdot \varepsilon_{\theta .n.m}$$
where $\varepsilon_{\theta .n}$ represents the measured circumferential strain of the steel ring “n” (m/m·10^−6^), and $\varepsilon_{\theta .n.m}$ represents the recorded circumferential strain of the steel ring “n”.

The calibration coefficient can also be used to rectify the concrete cracking time, as shown in Equation (5). In the case of a uniform deviation of recorded strains from all the strain gauges of a given ring, this clearly indicates stiffness that deviates from the stiffness of the theoretical ring. In such a situation, when the deformation deviation is in the range of 5% to 15%, it is reasonable to modify the recorded concrete cracking time with a calibration factor. Based on the results in Table 1, only one C-ring strain gauge read values significantly below the theoretical value, which clearly indicates the mounting error of this strain gauge and no reason to modify the cracking time for this ring.
(5)$$t_{crack.n}=\frac{t_{crack,n.m}}{\gamma_{c}}$$
where $t_{crack.n}$ represents the measured cracking time of the steel ring “n” after the calibration (days), and $t_{crack.n.m}$ represents the recorded cracking time of the steel ring “n” (days).

The use of such calibration is necessary for each measuring ring, which was prepared for susceptibility to cracking tests in accordance with the Standard ASTM C 1581/C 1581M–09a.

### 4.3. σ-ε Relation

The use of calibration coefficients for each measuring ring allows for a common interpretation of results, averaging the deformation values, determining of the average cracking time as a mean of the cracking times for individual samples, and determining the function of circumferential deformation of the measuring ring $\varepsilon_{\theta }$ to maximum values of circumferential stresses in concrete ring samples $\sigma_{\theta max,c}$. Figure 8 presents the linear relationship of the discussed parameters. 

## 5. Experimental Research

An analysis of the impact of the steel measuring ring calibration was carried out for two self-compacting concretes: concrete C-1 with fine and coarse natural aggregate, and concrete C-2 with pre-soaked fine and coarse lightweight aggregate. Two types of concrete shrinkage tests were performed for both concretes analyzed; the first was based on concrete deformation after 24 hours from concreting, while the second did not involve sample deformation. 

The composition of concrete mixes under consideration is shown in Table 3. Annular concrete samples were formed around the steel measuring rings, and their geometry was in agreement with the requirements of ASTM C 1581/C 1581M–09a. The measuring stands were placed in a climatic chamber where tests were carried out at a constant temperature T = 20 ± 2 °C and relative humidity RH = 50 ± 3%. The designed concretes were to have a high susceptibility to cracking under the influence of total shrinkage.

Deformation tests were carried out simultaneously on three calibrated measuring rings, as shown in Figure 9. 

Figure 10 and Figure 11 present the results of type 1 steel ring deformation tests and the development of tensile stresses on the inner surface of concrete samples from the moment of their formation, followed by deformation after 24 h, and until their cracking as a result of progressive drying shrinkage.

The performed deformation tests allowed to conduct two separate analyses. The first analysis concerned the deformations of the measuring ring C before and after calibration, taking into account the determined calibration factor. On its basis, it can be concluded that calibration validates the C ring relative to rings A and B. Therefore, the deformation values are characterized by a low standard deviation and allow for determination of the average deformation development affecting the correct interpretation of the results.

The second analysis referred to the interpretation of the material properties of concrete based on the relationship of the steel ring deformation to tensile stress on the inner surface of the annular concrete samples as a function of time. The natural aggregate used in C-1 concrete resulted in higher strength parameters as well as a more airtight and homogeneous structure compared to C-2 concrete with lightweight aggregate. Yet, C-1 concrete cracked in the third day after concreting at the average deformation value of the steel ring of −76.8 µm/m and a mean tensile stress of 6.2 MPa at the inner surface of concrete samples. The dynamic development of autogenous shrinkage in the first day and the additional impact of drying shrinkage after one day resulted in a rapid loss of strength due to the cracking of C-1 concrete samples. In the case of C-2 concrete, no autogenous shrinkage was observed in the first day and there was moderate development of the shrinkage from drying out after sample deforming. Light soaked aggregate led to internal care, which caused a slower development of shrinkage and stress. The use of lightweight aggregate extended the cracking time to about 5 days and reduced the strength of the concrete. C-2 concrete cracking occurred at the average deformation value of the steel ring of −16.3 µm/m, causing an average inner surface tensile stress of 1.3 MPa. Figure 12 shows the morphology of concrete sample cracks after the loss of strength due to autogenous and drying shrinkage. The development of deformation of the measuring rings reflects the homogeneity of the material structure. Hence, it is observed that for concrete C-2, the deformation course was more irregular.

In the following type 2 restrained concrete tests, the impact of steel rings calibration on the correctness of measurements over a longer period of time was analyzed. Ring samples of C-1 and C-2 concretes were not deformed after 1 day but remained insulated for 28 days. At that time, only autogenous shrinkage developed, and its impact on steel ring deformations was analyzed. The test results are shown in Figure 13 and Figure 14. 

The measurement of deformation of steel rings under the influence of autogenous shrinkage, especially for concrete C-1, showed the correctness of the calibration procedure in the range of 28 days. For the C-ring, the results before and after calibration are presented. The application of the calibration factor for the C-ring deformation course allowed for correct analysis of the results and the determination of concrete susceptibility to cracking in both short and long measurement periods.

Based on the analysis of the development of parameters of C-1 concrete with natural aggregate, a monotonic increase in the deformation of the measuring rings can be noticed as a result of the continuous development of autogenous shrinkage of concrete. Within 28 days, concrete does not show susceptibility to cracking at a given limitation level. On the other hand, the nature of the increase and the value of the average tensile stress at the inner surface of the concrete samples at the level of 5.5 MPa may indicate the development of micro-cracks in the structure and fracture of the samples at a later time. A lack of sample cracking within 28 days is caused by the increase in concrete strength during the test and by the absence of drying shrinkage.

Analysis of the deformation progress of measuring rings for C-2 concrete with lightweight aggregate showed no impact of autogenous shrinkage. In the whole measuring range, the soaked lightweight aggregate showed curing properties, as a result of which autogenous shrinkage did not develop in concrete ring samples. The registration of steel ring deformations throughout the entire measuring range was between 0 and −10 µm/m, generating minimal tensile stress in concrete samples.

Figure 13 proves the necessity of steel ring C calibration, where the calibrated strain values converge with strains for rings A and B. The uncalibrated, recorded strain of ring C was plotted as well and shows an approximate deviation of concrete tensile stresses after 28 days of about 0.4 MPa, which translates to underestimation of about 7% relative to mean stress from all the samples. 

## 6. Conclusions

In the calibration test, the pressure applied to the measuring ring by the rubber collar imitates the load caused by the shrinkage of concrete. Measurement of the air pressure with the digital manometer determines the graph of the circumferential strain–radial stress function. The calibration test additionally eliminates the error caused by the geometry and elastic modulus of the material. Pneumatic calibration allows compensating for errors due to improper strain gauge installation by the application of a test-determined calibration coefficient, which translates registered strains into calibrated circumferential strains closely aligned to theoretical values.

The calibration procedure allowed for simuntanous strain measurements under given stress for all rings of the test bench. Acquired calibration functions are used to calculate the mean values of the results, which can be used in further studies. Calibrated strains help determine the stresses occurring at the moment of cracking of concrete ring samples using a standard rigid measuring ring. This enables the classification of concrete’s susceptibility to cracking.

Short-term and long-term tests confirm the effectiveness of calibration to correctly interpret the test results on concrete susceptibility to cracking using restrictive rings. The applied calibration method extends the scope of tests with the correct analysis of the average deformability of steel rings and determination of the value of tensile stress in concrete at a given level of steel ring deformation.

In the future, it is planned to carry out research on the impact of the percentage of mineral additives on the time of cracking of concrete ring samples caused by the effect of autogenous shrinkage.

## 7. Patents

No. PL225785: Method for the calibration of measuring rings used for measuring their deformability in result of the shrinking strain of poured-in materials and the system for the calibration of measuring rings used for measuring their deformability as a result of the shrinking strain of poured-in materials.

## Figures and Tables

**Figure 1 materials-13-02963-f001:**
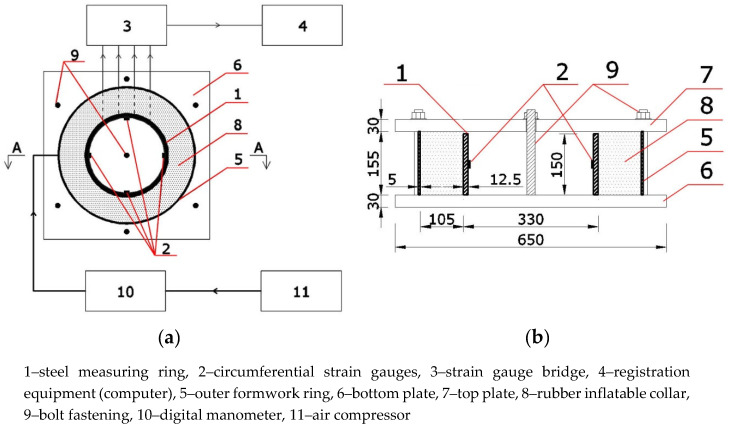
Block diagram of the calibration system for steel measuring rings: (**a**) top view; (**b**) section A-A.

**Figure 2 materials-13-02963-f002:**
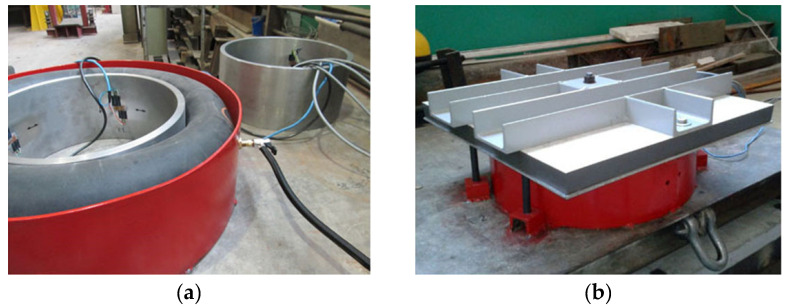
Calibration test bench: (**a**) placement of measuring ring, rubber collar, and shielding ring; (**b**) isolated with the top plate, bottom plate, and outer shielding ring.

**Figure 3 materials-13-02963-f003:**
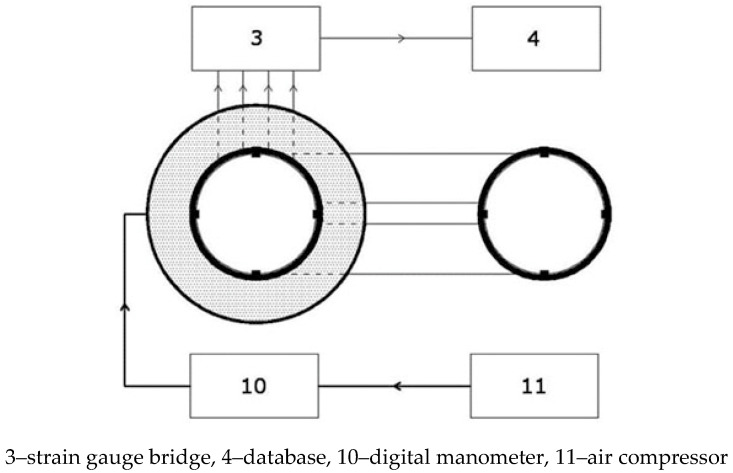
Calibration system utilizing a strain gauge bridge without internal temperature compensation: 4 pairs of strain gauges.

**Figure 4 materials-13-02963-f004:**
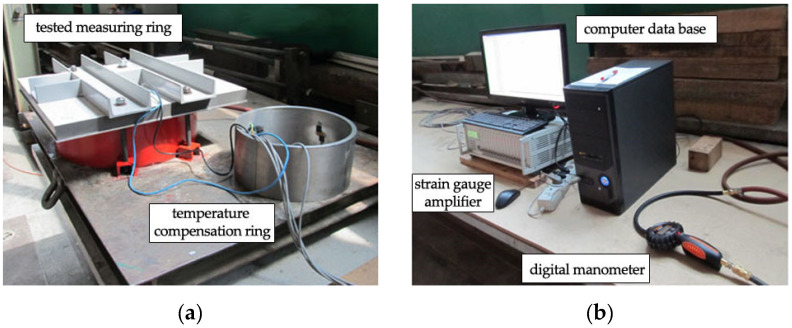
Components of the calibration system: (**a**) measuring rings during the test; (**b**) registration of the strain and air pressure.

**Figure 5 materials-13-02963-f005:**
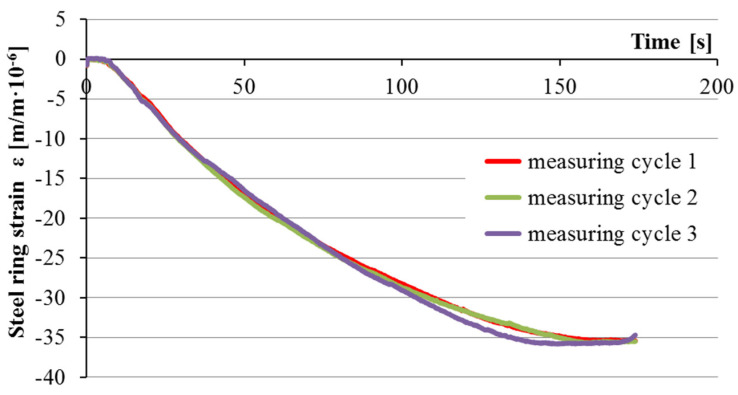
Steel ring B strains in relation to external pressure ranging from 0 to 5.5 bar.

**Figure 6 materials-13-02963-f006:**
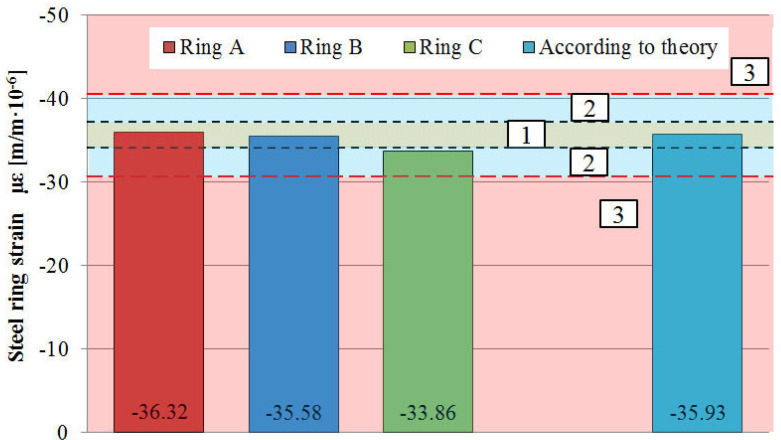
Circumferential deformation of tested steel rings under 5.5 bar pressure with allotment to measuring correctness zones: 1—deformation of the ring within ±5% tolerance, 2—deformation of the ring requires the use of a calibration factor within ±15%, 3—incorrect registration of ring deformation.

**Figure 7 materials-13-02963-f007:**
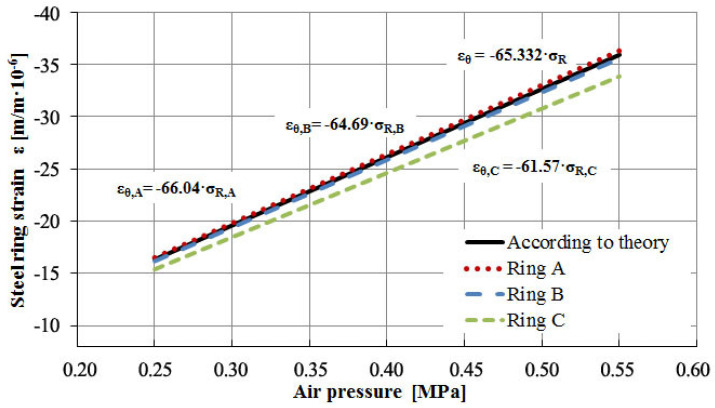
Determination of calibration coefficient for individual ring.

**Figure 8 materials-13-02963-f008:**
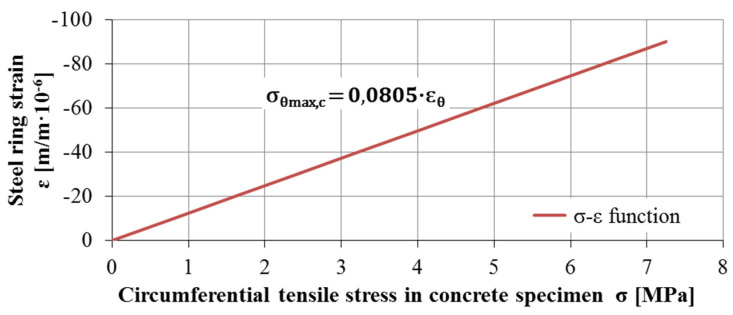
Theoretical relationship between steel ring strain and concrete ring sample tensile stress.

**Figure 9 materials-13-02963-f009:**
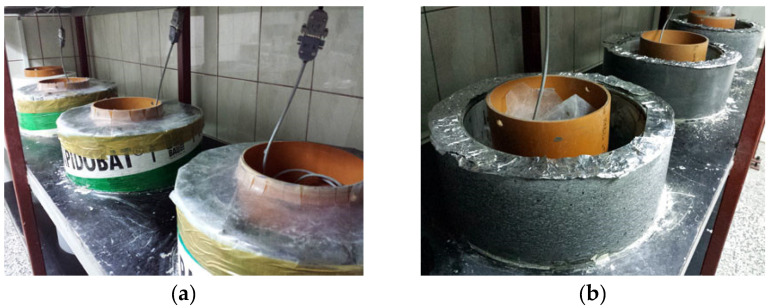
Restrained shrinkage test of concrete: (**a**) concrete samples insulated and subjected to autogenous shrinkage; (**b**) side formwork removal after 24 h of concreting and measurement of the impact of the drying shrinkage.

**Figure 10 materials-13-02963-f010:**
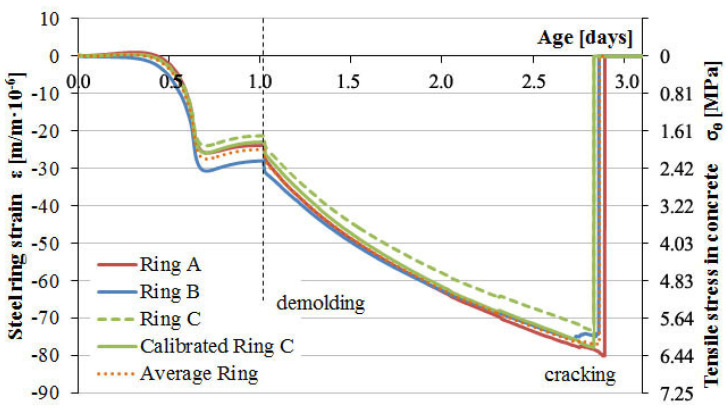
Strain development in the steel rings and stress progress in concrete C-1 induced by the total shrinkage.

**Figure 11 materials-13-02963-f011:**
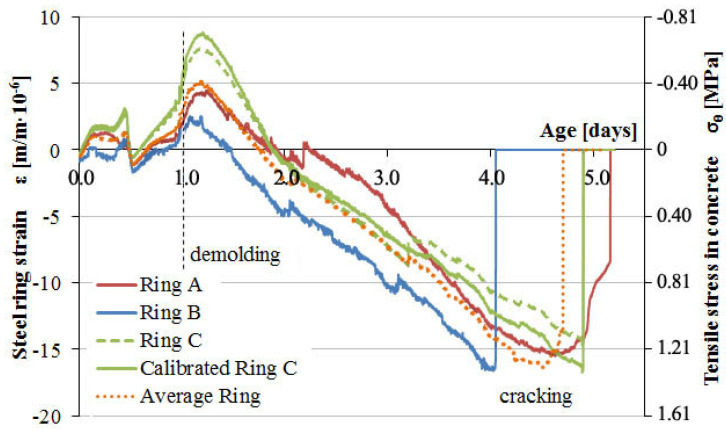
Strain development in the steel rings and stress progress in concrete C-2 induced by the total shrinkage.

**Figure 12 materials-13-02963-f012:**
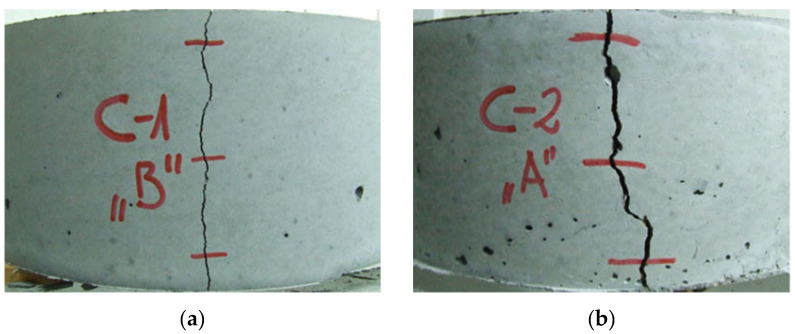
Cracked concrete ring specimens: (**a**) high-performance concrete with coarse natural aggregate 2-8, crack width = 0.9 mm; (**b**) high-performance concrete with coarse lightweight aggregate 4-8, crack width = 2.4 mm.

**Figure 13 materials-13-02963-f013:**
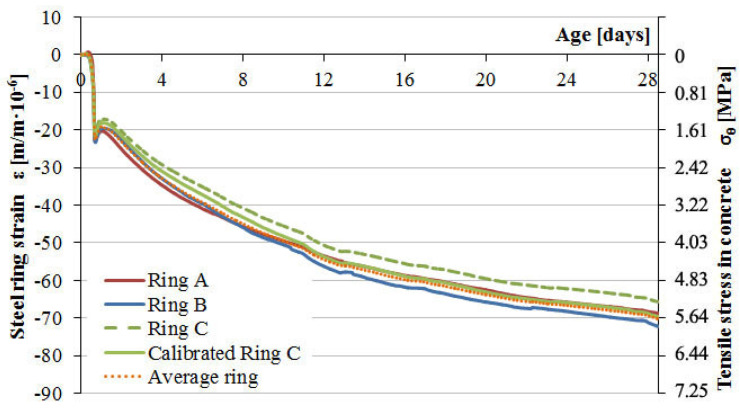
Strain development in the steel rings and stress progress in concrete C-1 induced by the autogenous shrinkage.

**Figure 14 materials-13-02963-f014:**
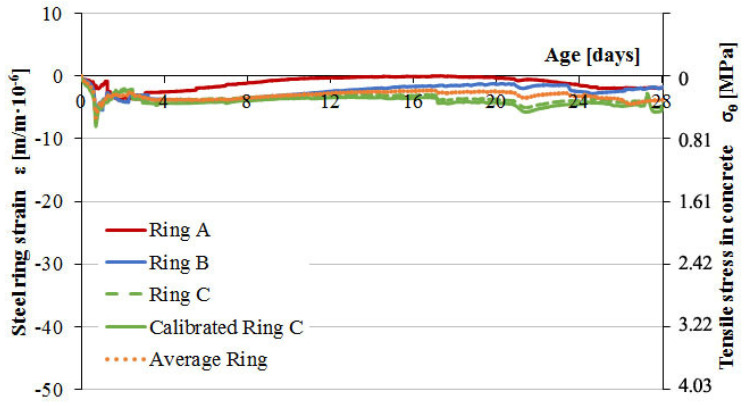
Strain development in the steel rings and stress progress in concrete C-2 induced by the autogenous shrinkage.

**Table 1 materials-13-02963-t001:** Measured circumferential strains for individual strain gauges under constant pressure (5.5 bars).

Measuring Cycle (MC)		1: Starting Position			2: 90° Turn		3: 180° Turn		Gauge Mean	Ring Mean/ Theoretical Ring	
Repetition		1 time		2 time	1 time	2 time	1 time	2 time			Deviation
**Ring A**	Strain per gauge [m/m·10^−6^]	1	−37.26	−36.98	−35.54	−37.01	−36.67	−36.21	−36.61	101.9%	1.9%
		2	−36.13	−36.11	−36.67	−36.23	−36.54	−36.04	−36.29	101.0%	1.0%
		3	−35.47	−36.14	−35.54	−36.16	−37.24	−36.54	−36.18	100.7%	0.7%
		4	−36.55	−36.07	−36.33	−36.31	−36.18	−35.73	−36.20	100.7%	0.7%
	Ring Mean per MC	−35.35		−36.33	−36.02	−36.43	−36.66	−36.13	**−36.32**	**101.1%**	**1.1%**
**Ring B**	Strain per gauge [m/m·10^−6^]	1	−35.82	−36.05	−35.25	−35.99	−35.71	−35.89	−35.79	99.6%	−0.4%
		2	−35.43	−35.64	−35.97	−35.16	−35.58	−35.16	−35.49	98.8%	−1.2%
		3	−35.36	−35.64	−35.92	−35.74	−34.84	−35.86	−35.56	99.0%	−1.0%
		4	−35.03	−36.04	−35.14	−35.81	−35.81	−35.11	−35.49	98.8%	−1.2%
	Ring Mean per MC	−35.41		−35.84	−35.57	−35.68	−35.49	−35.51	**−35.58**	**99.0%**	**−1.0%**
**Ring C**	Strain per gauge [m/m·10^−6^]	1	−34.98	−35.11	−35.57	−35.27	−35.78	−35.47	−35.36	98.4%	−1.6%
		2	−35.51	−34.14	−34.05	−34.52	−34.54	−34.68	−34.57	96.2%	−3.8%
		3	−30.57	−30.87	−31.21	−30.94	−30.56	−30.68	−30.81	85.7%	−14.3%
		4	−34.89	−34.71	−34.94	−34.48	−34.5	−34.64	−34.69	96.6%	−3.4%
	Ring Mean Per MC	−33.99		−33.71	−33.94	−33.8	−33.85	−33.87	**−33.86**	**94.2%**	**−5.8%**
**Theoretical ring**									**−35.93**	**100.0%**	**0.0%**

**Table 2 materials-13-02963-t002:** Error tolerance and calibration coefficients.

Ring	Strain at 5.5 Bars [m/m·10^−6^]	Deviation [%]	Calibration Coefficient [-]
A	−36.32	1.1	1.000
B	−35.58	−1.0	1.000
C	−33.86	−5.8	1.061
**Theory**	−35.93	-	-

**Table 3 materials-13-02963-t003:** Composition and notification of concrete mixes.

Concrete	Cement 42,5R [kg/m^3^]		Fly Ash [kg/m^3^]	Silica Fume [kg/m^3^]		Water [kg/m^3^]	SP [kg/m^3^]	Aggregate [kg/m^3^]			
								Natural		Lightweight	
								0–2	2–8	0–4	4–8
C1/450/NA	450	72			38	155	11	624	1072	-	-
C2/450/NA-LWA	450	72			38	155	7.65	-	-	310	540

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
