# Peer review of "Calibration of Steel Rings for the Measurement of Strain and Shrinkage Stress for Cement-Based Composites"

_materials, 2020, doi:10.3390/ma13132963_

Round 1

Reviewer 1 Report

The article "Calibration of steel rings for the measurement of strain and shrinkage stress for cement-based composites" is a complete, well presented and well explained work. I would suggest only some typographical changes (for points below) or colors of the figures, to make the explanation of the work even more effective.

  • In line 15 there is no space after the dot.
  • In figure 1 it would be possible to make the lines indicating the various parts of the system in red color?
  • In the conclusions it would be necessary to add a small part of future perspective on the use (even if not for research purposes) of this system.

Author Response

Thank you for your interest and reviewing my paper.

My reply to your comments:
- sentence: space added - "... time. Steel ..."
- the color of the indicator lines replaced in Fig. 1 - I confirm that the comment is correct
- I expanded the article's conclusions to include future scientific research.

In addition, I expanded the introduction.

I am attaching a revised version of the article in track change mode.

Reviewer 2 Report

This manuscript was very interesting.

I have some comments shown below.

  • The introduction is very weak compared to the rest of the parts.
  • The introduction must be improved to be published as a journal paper.
  • The literature review is incomplete. The authors did not provide all the previous works similar to their work in the introduction. This part should be improved.
  • The problem statement, research objective, the novelty of this study must be emphasized between "1. Introduction" and "Method and Experiment Program".
  • Sometimes, researchers measure the strain in the room temperature. Sometimes, researchers measure the strain in the high humidity and high temperatures. Thus, the authors need to provide the limitation of this proposed method.
  • Are there any effects by “steel rings” because Steel rings are susceptible to the temperature? It is possible that the volume changes of the steel rings will impact the strain measurement.
  • Lines 37-39: “In modern concretes with low water/cement ratio, the overall shrinkage is significantly affected by the autogenous shrinkage, which occurs in the first stage of hardening”. (Please, add a reference)
  • Lines 54-57: “Obtained steel ring deformation values and developed tensile stresses in annular concrete samples were analyzed for two maturation conditions: deformation due to autogenous and drying shrinkage - side formwork removed after 24 hours of concreting - and deformation due to autogenous shrinkage only without side surface drying effects.” (Need to add a justification why two maturation conditions the authors selected.)
  • Figure 1: Add the unit in the caption 
  • Line 74: “The rubber collar should be connected through a digital manometer to the air compressor.” (Please, add the reason.)
  • Figure 3: (Can you provide this Figure as a 3-dimensional Figure?)
  • Lines 108-109: “The procedure is recommended, especially for systems using temperature compensation through the vertical axis.” (Please, add the reason)
  • Conclusion: (Using some bullet points to summarize this study.)
  • Line 299: “Proper measurements” What is the meaning of “Proper”? (Please, use the scientific expression.)

Author Response

Thank you for your interest and reviewing my paper.

My reply to your comments:

- I agree with the comments on introduction. I have expanded the introduction. I described different research of restrained ring test along with extended bibliography.

- I have added an additional point 2. Research Problem. I described the purpose and novelty of the research in it. I expanded my conclusions with plans for future - research.

- The deformation of steel rings must be measured at a certain temperature. The thermal expansion of the material has an effect. Failure to take into account thermal parameters will result in misinterpretation of results. The error will concern the omission of the change in the volume of the measuring rings. Therefore, the calibration procedure and tests were carried out at a constant temperature of 20 Celcius degrees.

- Lines 37–39 (nowe linie: 56-60) rozszerzyłem akapit o "High-performance concretes undergo autogenous shrinkage even up to 200 µm / m after the first day of ripening. In the case of traditional concretes with a water / cement ratio of 0.5, the value of autogenic shrinkage after 28 days reaches 100 µm / m and in practical conditions is negligible" na podstawie Aïtcin, P.C. High Performance Concrete.

- Lines 54-57 (new lines 81-86): explanation of why tests were performed in two time modes

- Figure 1 - unit of measure added

- Line 74 (new lines 109-110): clarification added

- Fig. 3 - unfortunately, now I do not have a 3D design program

- Lines 108-109 (new lines 144-145) - the presented procedure was recommended for the calibration of steel rings connected to a strain gauge bridge without thermal compensation. Such recommendations were related to the equipment available in my laboratory. I agree, this is not an argument. The procedure is appropriate for strain gauges with and without temperature compensation. That's why I deleted this sentence.

- Line 299 (new lines 335) - I used a technical expression: Calibrated strains.

I am attaching a revised version of the article in track change mode.
